# Metabolomics during canine pregnancy and lactation

Sebastian P. Arlt[1,2]\*, Claudia Ottka[3], Hannes Lohi[3,4,5], Janna Hinderer[2], Julia Lüdeke[2], Elisabeth Müller[6], Corinna Weber[6], Barbara Kohn[7], Alexander Bartel[8]\*

1 Clinic of Reproductive Medicine, Vetsuisse Faculty, University of Zurich, Zurich, Switzerland, 2 Clinic for Animal Reproduction, Faculty of Veterinary Medicine, Freie Universitaet Berlin, Berlin, Germany, 3 PetBiomics Ltd, Helsinki, Finland, 4 Department of Veterinary Biosciences and Department of Medical and Clinical Genetics, University of Helsinki, Helsinki, Finland, 5 Folkhälsan Research Center, Helsinki, Finland, 6 Laboklin GmbH & Co KG, Bad Kissingen, Germany, 7 Clinic for Small Animals, Faculty of Veterinary Medicine, Freie Universitaet Berlin, Berlin, Germany, 8 Institute for Veterinary Epidemiology and Biostatistics, Faculty of Veterinary Medicine, Freie Universitaet Berlin, Berlin, Germany

\* sebastian.arlt@uzh.ch (SPA); alexander.bartel@fu-berlin.de (AB)

**Data Availability Statement:** All relevant data are within the paper and its Supporting information files.

**Funding:** The costs were covered by the Freie Universitaet Berlin (examination, sampling) without

## Abstract

During pregnancy and parturition, female dogs have to cope with various challenges such as providing nutrients for the growth of the fetuses, hormonal changes, whelping, nursing, milk production, and uterine involution. Metabolomic research has been used to characterize the influence of several factors on metabolism such as inter- and intra-individual factors, feeding, aging, inter-breed differences, drug action, behavior, exercise, genetic factors, neuter status, and pathologic processes. Aim of this study was to identify metabolites showing specific changes in blood serum at the different phases of pregnancy and lactation. In total, 27 privately owned female dogs of 21 different breeds were sampled at six time points: during heat, in early, mid and late pregnancy, at the suspected peak of lactation and after weaning. A validated and highly automated canine-specific NMR metabolomics technology was utilized to quantitate 123 measurands. It was evaluated which metabolite concentrations showed significant changes between the different time points. Metabolites were then grouped into five clusters based on concentration patterns and biochemical relationships between the metabolites: high in mid-pregnancy, low in mid-pregnancy, high in late pregnancy, high in lactation, and low in lactation. Several metabolites such as albumin, glycoprotein acetyls, fatty acids, lipoproteins, glucose, and some amino acids show similar patterns during pregnancy and lactation as shown in humans. The patterns of some other parameters such as branched-chain amino acids, alanine and histidine seem to differ between these species. For most metabolites, it is yet unstudied whether the observed changes arise from modified resorption from the intestines, modified production, or metabolism in the maternal or fetal tissues. Hence, further species-specific metabolomic research may support a broader understanding of the physiological changes caused by pregnancy that are likely to be key for the normal fetal growth and development. Our findings provide a baseline of normal metabolic changes during healthy canine pregnancy and parturition. Combined with future metabolomics findings, they may help monitor vital functions of pre-, intra-, and post-partum bitches and may allow early detection of illness.

any specific funding and PetBiomics Ltd provided material support (Analyses). PetBiomics Ltd employee Claudia Ottka and PetBiomics Ltd chairman Hannes Lohi were involved in the analysis and the preparation of the manuscript. The funders had no role in study design, data collection and decision to publish.

**Competing interests:** I have read the journal's policy and the authors of this manuscript have the following competing interests: Claudia Ottka was an employee and Hannes Lohi is a shareholder and the chairman of the board of PetBIOMICS Ltd, who developed and provides the metabolomics test. All other authors declare to have no competing interests. This does not alter our adherence to PLOS ONE policies on sharing data and materials.

## Introduction

Pregnancy and parturition are challenging for mammals because they are characterized by considerable hormonal and metabolic changes [1, 2] to support fetal demands [3, 4]. During pregnancy the maternal body needs to supply appropriate nutrients for the conceptuses via the placentas and later the newborn puppies rely on lactation for several weeks [5]. These processes are governed by complex hormonal and neurological interactions and cause major alterations in the mothers metabolism [5], resulting in modified nutritional demands for both micro- and macronutrients [6, 7]. It has been claimed that even small deviations of the metabolism from the norm during pregnancy could have serious consequences for the mother and the offspring [8].

In dogs, pregnancy-related changes in metabolic parameters might be even more intensive than in many other mammalian species because fetal development is rapid as pregnancy duration is, on average, only 63 days, and a bitch might carry a high number of conceptuses. Parturition usually lasts several hours, is laborious, and often exhausting [9]. Other parameters may change during pregnancy and parturition because of hormonal changes, uterine enlargement, pain, anxiety and uterine contractions [10]. In addition, the canine endotheliochorial placentation type leads to a longer and more intensive uterine involution process than in other species [11]. Finally, also tasks such as nursing the puppies, including milk production and change of diet, are challenging for the female dog [1].

Understanding the physiology, endocrinology, and metabolic features during pregnancy, parturition, and lactation are required to understand and diagnose normal and abnormal conditions and provide optimal supportive management [1, 5]. In that regard, it is important to realize that some parameters change physiologically during pregnancy, parturition, and lactation [10]. However, it has been shown that inadequate supply of key metabolites can lead to abnormal fetal growth and impaired development of organs such as the heart [12] or nerval tissue [13].

Metabolomics is an omics-based approach that generates comprehensive information about metabolites in blood serum, enabling an extensive view of the individual's current state of systemic metabolism [14, 15]. Several metabolomics studies have been conducted in humans to address the physiological and pathophysiological metabolic changes during pregnancy and lactation, but no previous studies have been conducted in dogs. This study aimed to determine the metabolic changes occurring in dogs during pregnancy and lactation, to discuss their similarities or differences to the changes occurring in humans.

## Materials and methods

### Animals

In total, 27 privately owned bitches of 21 different breeds were sampled during this study (Table 1). Female dogs were enrolled from March 2018 until December 2019. To be included, bitches had to be generally healthy within the past 6 months before mating and not receive medication, except deworming. In addition, dogs had to become pregnant and be presented for sampling at six time-points. Previous gynecological illnesses were inquired about, but none were present in any of the bitches at the time of sampling. Informed written consent and approval from the owners was obtained. The dogs were fed individual diets chosen by the owners. The blood sample collection was approved by the committee on animal welfare of the federal state Berlin (LAGeSo Berlin O 0095/18).

Bitches were initially presented for ovulation timing. After they met the inclusion criteria, an initial clinical and gynecological examination was performed, including vaginoscopy and vaginal cytology.

**Table 1. Attributes of female dogs included in a study on metabolomics during canine pregnancy and lactation.**

| Dog No | Breed | Bodyweight (kg) | Parity* | Age | Number of puppies born | Thereof stillborn puppies |
|---|---|---|---|---|---|---|
| 1 | Dogue de Bordeaux | 51.3 | 4 | 7.0 | 2 | 0 |
| 2 | Miniature Bull Terrier | 16.8 | 1 | 2.9 | 6** | 3 |
| 3 | Rhodesian Ridgeback | 38.0 | 2 | 4.9 | 12 | 1 |
| 4 | Saarloos Wolfhound | 28.8 | 2 | 6.2 | 5 | 0 |
| 5 | Collie | 24.0 | 2 | 6.3 | 5 | 0 |
| 6 | Borzoi | 36.4 | 1 | 8.1 | 8** | 0 |
| 7 | Labrador Retriever | 22.4 | 1 | 2.6 | 9 | 0 |
| 8 | Staffordshire Bull Terrier | 14.2 | 1 | 2.6 | 1 | 0 |
| 9 | French Bulldog | 11.8 | 2 | 2.6 | 3 | 1 |
| 10 | Schnauzer | 18.2 | 2 | 7.8 | 6 | 0 |
| 11 | Olde English Bulldogge | 32.2 | 1 | 2.0 | 12 | 1 |
| 12 | Miniature Bull Terrier | 19.0 | 2 | 4.7 | 7** | 0 |
| 13 | Wirehaired Dachshund | 9.0 | 2 | 4.9 | 7 | 0 |
| 14 | Boxer | 27.8 | 2 | 6.8 | 4 | 0 |
| 15 | Hovawart | 41.6 | 2 | 6.7 | 9 | 1 |
| 16 | Irish Wolfhound | 56.4 | 1 | 2.9 | 13 | 1 |
| 17 | Golden Retriever | 22.8 | 1 | 4.4 | 5 | 0 |
| 18 | Entlebucher Mountain Dog | 23.6 | 1 | 2.6 | 7 | 0 |
| 19 | Golden Retriever | 33.0 | 1 | 4.6 | 6 | 0 |
| 20 | Miniature Bull Terrier | 16.0 | 2 | 5.4 | 6 | 1 |
| 21 | Eurasier | 20.8 | 2 | 3.8 | 5 | 0 |
| 22 | Wirehaired Dachshund | 9.2 | 1 | 3.4 | 6 | 0 |
| 23 | Kromfohrlaender | 10.4 | 1 | 2.9 | 7 | 1 |
| 24 | English Cocker Spaniel | 11.2 | 2 | 2.9 | 8 | 0 |
| 25 | Labrador Retriever | 28.2 | 2 | 5.3 | 6 | 0 |
| 26 | English Cocker Spaniel | 11.2 | 2 | 2.9 | 6 | 0 |
| 27 | Dobermann | 38.8 | 1 | 3.4 | 13 | 0 |
| | **Total (Average)** | **24.9** | **1.6** | **4.5** | **6.8** | **0.37** |

* including the current pregnancy,

** these dogs underwent C-section

### Blood sampling and sample processing

For this study, three stages of canine pregnancy were defined according to Hinderer et al. [16]. A total of six appointments for drawing blood samples were scheduled. The first sample was taken at a visit for ovulation timing in estrus (E). Three samplings during pregnancy (P) were appointed, i. e., between day 11 and 19 (P1) after mating, between day 23 and day 32 (P2), and between day 52 and 60 (P3). The P2 examination included an ultrasonic examination to assert whether the bitch was pregnant. This examination was not scheduled in the middle of the second third to respect the owners' wishes of having a pregnancy diagnosis rather sooner than later while completing the study examination within the same appointment. A fifth sampling was performed 18 to 24 days after whelping. The last clinical examination (A) was performed 18 to 24 days after the weaning of the litter.

All blood samples were collected in the morning or early afternoon.

After parturition, data on the delivery (natural or cesarean section), the number and weight of puppies (alive and stillborn), and survival after three weeks were documented.

Blood samples were collected by venipuncture from either the cephalic or the saphenous vein into plastic tubes (Sarstedt Tube 4.5 mL, Clotting Activator/Serum, 75 × 13 mm, Sarstedt AG & Co KG, Nümbrecht, Germany). Samples were left at room temperature for 30 to 60 min for clotting and then centrifuged at 5000 rpm or 2884 g for 5 min (Hettich Centrifuge EBA 20, Hettich, Tuttlingen, Germany). Serum was transferred into a serum tube (Simport Cryovial sterile with lip seal design, external threads, 2 mL Tubes, Boloeil QC, Canada), which was placed in a refrigerator at– 80˚C until shipment. The storage temperature was checked daily, and variations of the measured temperature remained between -80˚C and– 83˚C. Storage duration at– 80˚C of the samples ranged from 11 to 30 months.

Only complete sample sets with sufficient serum and no haemolytic appearance were submitted for Nuclear magnetic resonance (NMR) analyses.

For shipment preparation, the tubes were thawed for one to three hours, and then an amount of 0.3ml was transferred into a second serum tube (Simport Cryovial sterile with lip seal design, external threads, 2 mL tubes, Boloeil QC, Canada), both aliquots were again immediately frozen at– 80˚C. Overnight shipment to PetBiomics, Finland, was performed in a Styrofoam box with plenty of dry ice.

## Nuclear magnetic resonance (NMR) metabolomics analyses

A validated and highly automated canine-specific NMR metabolomics technology was utilized for metabolomics analyses of the collected serum samples [14]. The method quantitates 123 measurands from various molecular groups, including a comprehensive lipoprotein analysis, fatty acids, triglycerides, cholesterols, amino acids, glycolysis- and fluid balance-related metabolites, and a novel inflammatory marker glycoprotein acetyls (GlycA) [14]. A similar method has been largely utilized in human metabolomics studies [17–19], including a study on the metabolic effects of human pregnancy [2]. Technical details of the method are provided elsewhere [17–19].

## Statistical analysis

All statistical analyses were performed using R version 4.1.3 (R Foundation for Statistical Computing, Vienna). First, the data were checked for missing observations. The data did not include any metabolite values missing at random. Values below the detection limit were treated as zero values in the following statistical analyses.

For metabolite selection, a Friedman-Test (a non-parametric alternative to a repeated-measures ANOVA) was used to determine which metabolite concentrations showed significant changes between the different time points and thus are affected by the course of the pregnancy. The Friedman-Test allows taking into account the repeated measures of each animal (6 time points per animal), the corresponding null hypothesis is that there are no changes in metabolite levels within each animal over the course of the pregnancy. Based on the observed data, the normality assumption could not be reliably checked for every of the 123 parameters. Therefore, a non-parametric approach was chosen for its higher robustness. All p-values were corrected for multiple comparisons using the Benjamini-Hochberg method, and the p-value cutoff for adjusted p-values was set at $p < 0.05$.

Since p-values have limited interpretation for 'omics data, where the number of observations is much smaller than the number of variables (27 animals $<<$ 123 metabolites), we used $k$-means Clustering for dimensionality reduction [19]. $k$-means Clustering was done using the z-scaled values of each metabolite. The optimal number of clusters was determined using the "gap" statistic (R package "cluster" version 2.1.3, [20]). Cluster were afterward modified based on the biochemical relationships between the metabolites (super/subcategories, absolute/

relative values of the same metabolite), effectively creating 5 clusters from 123 variables. We assume that similar changes in the metabolites, defined by the clusters, are due to shared or related biological processes. To visualize the effect of the time points on the metabolites, "locally estimated scatterplot smoothing" (LOESS) was used. The reference intervals of the analysis [9] were added to the plots to provide reference on the typical concentrations of the metabolites and to highlight the magnitude of the change.

## Results

The 27 dogs enrolled in the study (Table 1) belonged to 21 different breeds and had a median body weight of 22.8 kg (Min: 9.0, Q1: 15.1, Q3: 32.6, Max: 56.4kg). The median age was 4.35 years (Min 1.98, Q1: 2.9, Q3: 5.77, Max 8.1 years).

Fig 1 shows the metabolic parameters which were high in mid-pregnancy (P2). Metabolites which showed different patterns, i. e. low in mid-pregnancy, high in late pregnancy, high in lactation, and low in lactation are presented in Fig 2 (cluster 2, 3,4,5).

## Discussion

Metabolism during pregnancy and lactation is a dynamic and precisely programmed process [8], as this stage of life is defined by rapid fetal or neonatal growth and development [2, 21]. Metabolite adaptations, such as increased hepatic glucose output, decreased peripheral insulin sensitivity, unchanged hepatic insulin sensitivity and lower peripheral insulin levels, allow the dam to maintain euglycemia during pregnancy [7, 22] and to support fetal growth despite the increasing nutritional demands [7]. Nevertheless, nutrition should be adapted to the respective pregnancy phase to ensure the health of the dam and the puppies [6]. It has been shown that protein deposition in maternal and fetal tissues increases throughout pregnancy, most occurring during the third trimester [21]. Furthermore, leptin increased proportionally with increased food intake during pregnancy, although an apparent body weight gain was observed only at day 60 [23]. The concentrations of metabolites in the blood serum of the bitch affect the corresponding concentrations and the composition of amniotic and allantoic fluid [7]. These fluids are essential during pregnancy, providing metabolites, protection, and the environment for normal fetal development [24].

Five distinct clusters were observed in the analysis of the patterns of the assessed parameters. The identified clusters were: parameters high (Cluster 1/Fig 1)/low (Cluster 2/Fig 2) in mid-pregnancy, parameters high at late pregnancy (Cluster 3/Fig 2), and parameters high (Cluster 4/Fig 2)/low (Cluster 5/Fig 2) in lactation. The clustering was based on similar patterns in metabolite concentrations with elevations and decreases in metabolite concentration occurring during similar time points within each cluster.

### Cluster 1: Parameters high in mid-pregnancy

**Total fatty acids** tended to be higher during pregnancy than at other time points, albeit the differences were not significant. However, an increase in the molar concentration of individual fatty acids was also observed, which may explain changes in the total fatty acids. In addition, elevated total fatty acids have also been observed in another study on dogs [5] and in human pregnancies because a high concentration of fatty acids is necessary to meet the demands of fetuses [3]. Fatty acids play an important role in the development of the fetus and may also be involved in modulatory effects on the mother's immune system [25]. It is well known that omega 3 fatty acids have anti-inflammatory effects [26] and it has been shown that some that fatty acids such as DHA can increase so called resolvins, which act inflammation resolving and anti-inflammatory [27]. Another strong link between metabolism and the immune system is

## Cluster 1: parameters high in mid pregnancy

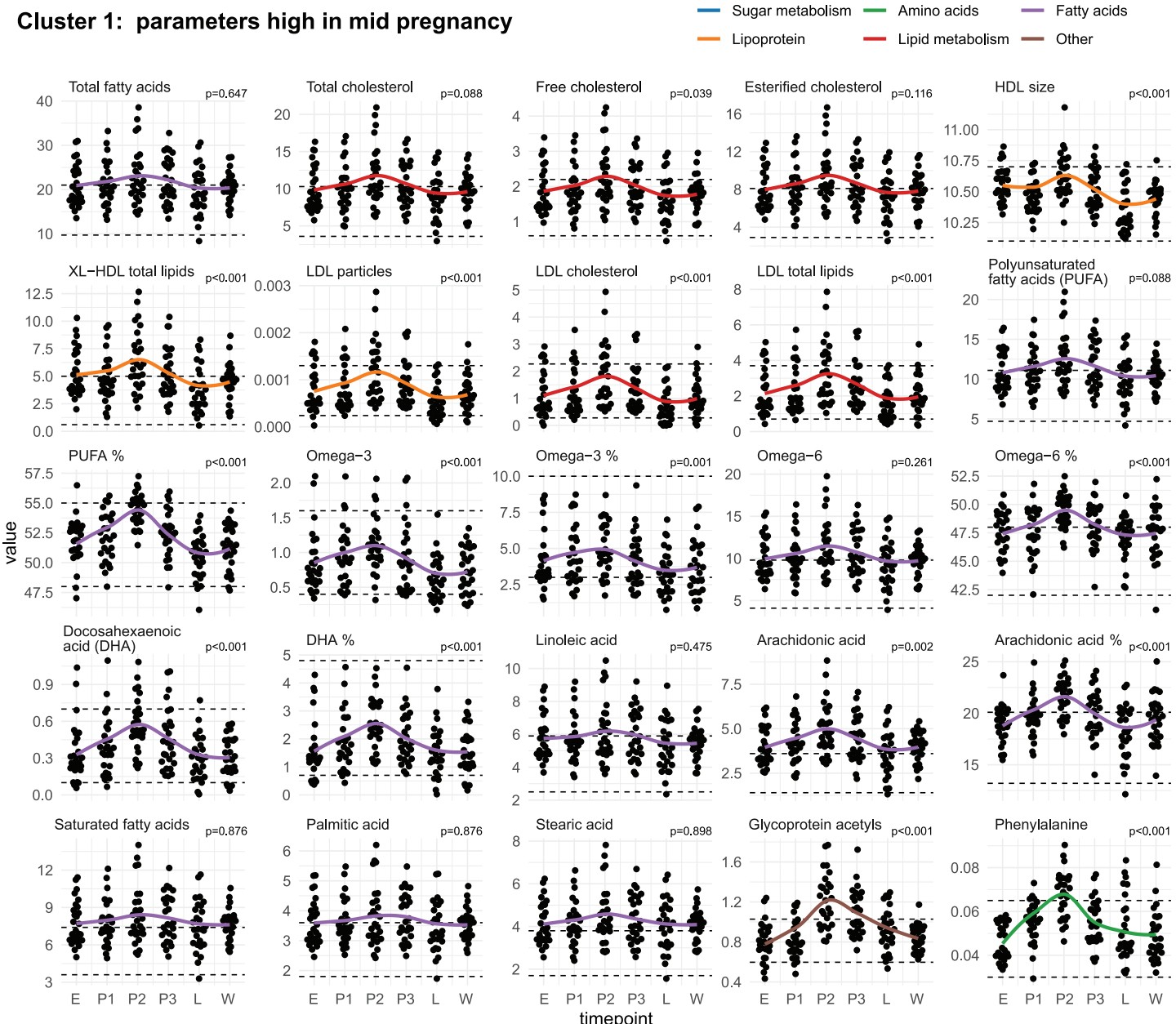

**Fig 1. Metabolic parameters high in mid-pregnancy (Cluster 1).** Change in metabolic parameters over the course of canine pregnancy (E = estrus, P1-3 = early/mid/late pregnancy, L = lactation, A = after weaning). Points show the individual measurements. The solid line shows the change in average metabolite levels (LOESS). Dashed lines show the upper and lower limits of the metabolite reference intervals [9]. The p-values were calculated using a Friedman-Test with Benjamini-Hochberg correction. The p-value cutoff for adjusted p-values was p< 0.05.

the fatty acid composition of membranes of immune cells affects phagocytosis capabilities, T cell signaling and antigen presentation capability [28].

The concentration of **free cholesterol (significant)** was highest in mid-pregnancy. The same pattern was mirrored in **total cholesterol** and **esterified cholesterol** (both not significant). These findings are not following another study assessing 12 dogs assigned to four different diet groups in which plasma total cholesterol concentrations were depressed in early

## Cluster 2: parameters low in mid pregnancy

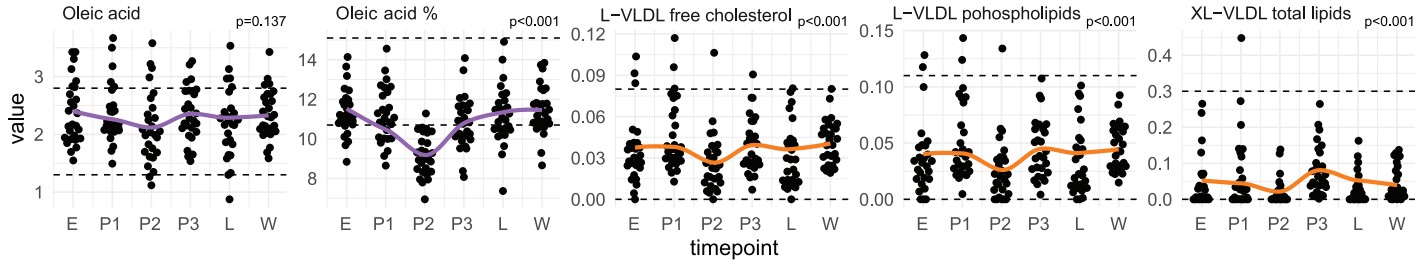

## Cluster 3: parameters high at late pregnancy

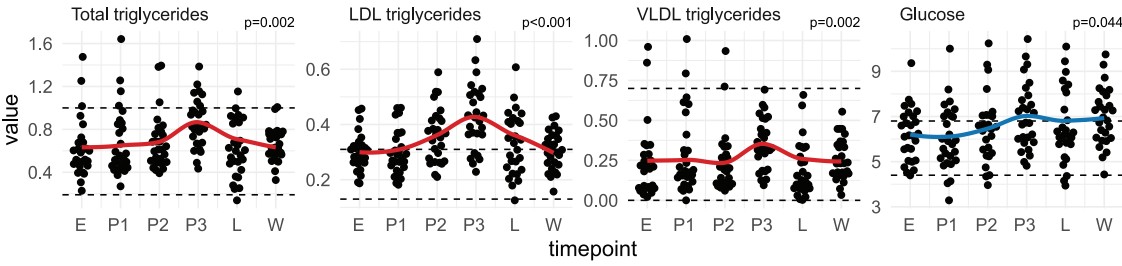

## Cluster 4:  parameters high in lactation

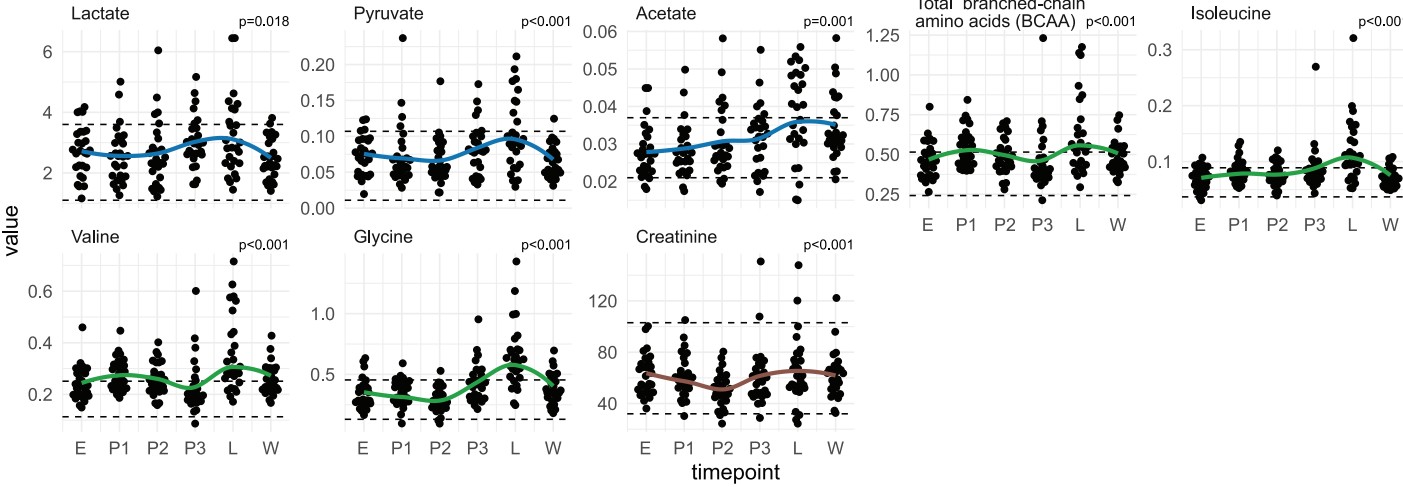

## Cluster 5: parameters low in lactation

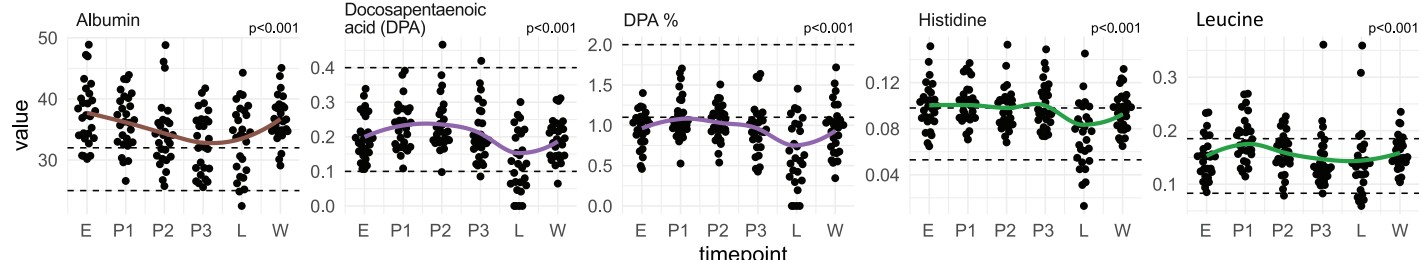

**Fig 2. Metabolic parameters low in mid-pregnancy, high in late pregnancy, high in lactation, and low in lactation (Cluster 2,3,4,5).** Change in metabolic parameters over the course of canine pregnancy (E = estrus, P1-3 = early/mid/late pregnancy, L = lactation, A = after weaning). Points show the individual measurements. The solid line shows the change in average metabolite levels (LOESS). Dashed lines show the upper and lower limits of the metabolite reference intervals [9]. The p-values were calculated using a Friedman-Test with Benjamini-Hochberg correction. The p-value cutoff for adjusted p-values was p< 0.05.

gestation and then increased in the later stages independently of the diet [29]. These differences refer mostly to a sample taken on day 42. At this time point, we did not sample the dogs. In another study no changes were found [5]. However, samples were taken at various and not exactly specified time points in pregnancy. It is well known that cholesterol also increases during human pregnancies [30].

Lipoproteins are divided into four major classes: chylomicrons, very low-density lipoproteins (VLDL), low-density lipoproteins (LDL), and high-density lipoproteins (HDL), with HDL being the most abundant lipoprotein in dogs [31]. Chylomicrons and VLDL are the main carriers of serum triglycerides, while HDL and LDL contain mainly cholesterol [32].

Our results show that the lipoprotein parameters show different patterns in the course of pregnancy and lactation in the dog. While the concentration of HDL particles (HDL-P) and VLDL particles (VLDL-P) show no changes, many others, such as **HDL size**, **total lipids in very large HDL (XL-HDL total lipids)**, the **concentration of LDL particles (LDL particles), LDL cholesterol, total lipids in LDL (LDL total lipids)** peak in mid-pregnancy and decline after that. Other lipoprotein parameters peak in late pregnancy (see cluster 3). An increase of lipoprotein concentrations during canine pregnancy has also been shown in other studies [29]. It is well known that dogs have a unique lipid metabolism and produce substantial, cholesterol-rich HDL particles in hypercholesterolemia. Other than in dogs, in humans, these cholesterols are usually transformed into the more atherogenic LDL particles, which is believed to increase the risk of atherosclerosis in humans [32].

The percentage of polyunsaturated fatty acids (**PUFA %**) (significant) and **PUFA** (not significant) were high in mid-pregnancy. The same phenomenon has been shown in humans [3, 25]. In humans, polyunsaturated fatty acids (PUFA) accumulate in maternal fat depots in early pregnancy and become available for placental transfer during late pregnancy when the fetal growth rate is maximal and fetal requirements for PUFAs are greatly enhanced [33]. A high demand for PUFAs has also been shown in dogs [13].

**Omega-3** unsaturated fatty acids and **omega-3%** were high in mid-pregnancy (both significant), as well as **omega-6** (not significant) and **omega-6%** (significant). Omega-3 and Omega-6 PUFA are needed to ensure the physiologic development of nervous tissue, retinal tissue, kidney, liver, and skin functions [29].

**Docosahexaenoic acid (DHA)** belongs to Omega-3 fatty acids and was significantly elevated in mid-pregnancy, as well as **DHA%**. It plays a significant role during pregnancy and lactation because it contributes to brain development, accounting for over 10% of brain fatty acids [34]. DHA increases during gestation in humans, but its proportion of total lipids decreases [35]. The authors suggest that this indicates a preferential transfer of DHA across the placenta to the fetus [35].

Arachidonic acid (AA), an Omega-6 fatty acid, and AA% were significantly elevated in mid-pregnancy. At the same time, **linoleic acid (LA)**, also an Omega-3 fatty acid and precursor for DHA and AA, was only slightly elevated in dogs. Concentrations of LA and AA also increase in pregnant women [35].

**Total saturated fatty acids** (**SFA**), including **palmitic acid (PalA)** and **stearic acid (SteA)**, were slightly elevated during pregnancy in dogs. In humans, the increase of SFA and PalA is markedly higher until the third trimester [25].

**Glycoprotein acetyls (GlycA)** is a novel spectroscopic marker of systemic inflammation combining the *N*-acetyl methyl group proton signals of several acute-phase proteins. It peaks significantly in mid-pregnancy in dogs, which is in accordance with other recent findings [5]. In humans, the intra-individual biological variation of this marker is low, and the main acute-phase protein contributors to its signal are α1-acid glycoprotein, haptoglobin, α1-antitrypsin, and α1-antichymotrypsin [36]. It has been shown that GlycA is higher in pregnant women

than in non-pregnant women [3]. The placentation process might explain the rise of this parameter and peak during mid-pregnancy. It is well known that acute phase proteins such as fibrinogen and c-reactive protein increase in the second half of pregnancy [37, 38]. A rise of acute phase proteins during pregnancy has also been shown in 12 pregnant bitches [39].

These findings follow the research on eight pregnant Beagle dogs, which has shown that Alpha-1-acid glycoprotein, which contributes to the GlycA signal, rises during dog pregnancy and peaks at around 45 days of gestation" [40]. In humans, Alpha-1-antitrypsin, another contributor to the GlycA signal, increases in pregnancy [41].

The only amino acid which shows a marked peak during mid-pregnancy is **phenylalanine**. In humans, it has been demonstrated that phenylalanine, alanine, and histidine were increased in pregnant women [3]. In dogs, alanine and histidine seem not to increase during pregnancy. Phenylalanine is required for protein synthesis and is intracellularly converted to tyrosine, which, in turn, is either used for protein synthesis, oxidized, or converted to the important neurotransmitters epinephrine, norepinephrine, and dopamine [42]. It has been shown that excessive feeding of phenylalanine to rats impairs proliferation and hypertrophy in the brain of their fetuses [43], but the dietary requirement for phenylalanine in healthy human pregnancies or canine pregnancies has not been determined yet. Dietary phenylalanine requirements seem not to vary between adult and non-pregnant dog breeds [44].

## Cluster 2: Parameters low in mid-pregnancy

**Oleic acid** is a monounsaturated omega-9 fatty acid. Compared to other unsaturated fatty acids (see Cluster 1), its concentration slightly declines in mid-pregnancy. In humans, Oleic acid increases during pregnancy [25].

**Free cholesterol** and **phospholipids** in large VLDL (L-VLDL) and **total lipids in extensive VLDL (XL-VLDL total lipids)** decline significantly during mid-pregnancy and increase towards the end of pregnancy. An increase in all parameters during pregnancy has been observed in humans [45].

## Cluster 3: Parameters high in late pregnancy

Through various mechanisms, pregnancy causes insulin resistance, suppressing the intracellular transport of glucose and increasing blood glucose concentrations in dogs [4, 46, 47]. Glucose concentrations significantly increase during late pregnancy and remain high until the time after weaning. Despite the pronounced insulin resistance in pregnant compared with non-pregnant diestrous bitches [38], gestational diabetes mellitus in dogs is rare [48]. The same mechanisms of insulin resistance have been described in humans [49, 50], albeit several groups have found lower glucose concentrations in pregnant women compared with non-pregnant ones [3]. The mechanism causing this decline is not well understood. Potential contributing factors include dilutional effects, an increased utilization by the fetoplacental unit or increased maternal uptake secondary to increased β-cell function, and/or inadequate hepatic production [50].

**Total triglycerides, LDL Triglycerides, and VLDL triglycerides significantly** peak in late pregnancy. This is in accordance with findings in normal human pregnancies. Shen et al. [51] found that the levels of lipids, including TG, TC, and LDL cholesterol, increased gradually during gestation and peaked before delivery; meanwhile, HDL cholesterol amounts increased from the 1st to 2nd trimester with a slight decrease in the 3rd trimester. Furthermore, in human pregnancies, an abnormal increase in triglycerides during pregnancy is associated with gestational diabetes mellitus [51]. The possible disease correlations of abnormally high triglycerides during canine pregnancy require further study. Similar to gestational diabetes mellitus,

hypertensive disorders are not common complications of canine pregnancy. Interestingly, unlike humans, dogs' HDL triglycerides are not affected by pregnancy or lactation.

## Cluster 4: Parameters high in lactation

It has been shown that lactate and pyruvate are more elevated in pregnant women than in non-pregnant women [3]. The **lactate, pyruvate**, and **acetate** concentrations significantly rise towards late pregnancy and lactation, presumably reflecting the higher demands of energy metabolism, even if their medians remain within the reference intervals at all time points. Whether lactate can be used as an indicator for sepsis or other pregnancy-related disorders in dogs as it is used in humans [52] requires further study since all bitches included in our study had undisturbed pregnancies.

The total concentration of **branched-chain amino acids (BCAA)**, which includes leucine, isoleucine, and valine, shows a marked increase during lactation. Interestingly, isoleucine and valine show the same pattern, whereas leucine shows low concentrations during lactation (see cluster 5). The same is true for the amino acid **glycine**. Glutamine, glycine, valine, and tyrosine were lower in pregnant women compared with non-pregnant women; no differences were found for isoleucine and leucine between pregnant and non-pregnant women [3].

**Creatinine** shows a significant decrease in mid-pregnancy, which is in accordance with other findings in dogs [5] and in pregnant women. It has been stated that a physiologic increase in the glomerular filtration rate during pregnancy results in a decrease in the concentration of serum creatinine [53]. In addition, it has been shown that creatinine was lower in pregnant women compared to non-pregnant women [3].

## Cluster 5: Parameters low in lactation

The observed significant decline of **albumin** during late pregnancy and lactation has also been well documented in dogs [5] and humans during and after normal pregnancies [3, 54]. It is believed that this decline is caused by physiologic hemodilution but may lead to physiological edema and even may associate with conditions such as eclampsia [54]. In addition, it has been shown that albumin concentrations increase in the allantoic fluid in late canine pregnancy [24].

**Docosapentaenoic acid (DPA)** is also significantly elevated during pregnancy but shows a significant decline during lactation, which is also visible using the parameter **DPA%**.

The concentration of the amino acid **histidine** remains stable at all time points, except for a significant decrease during lactation. Histidine likely plays a role in milk production, as it has been shown for cows, who have an increased energy-corrected milk yield when supplemented with histidine [55].

The amino acid **leucine** showed a significant increase in its concentration during early pregnancy and a decline during mid and late pregnancy and lactation. This contrasts with other branched-chain amino acids (see cluster 4). This decline is likely connected with milk production since leucine plays a vital role in milk production, as shown in cows [56].

## Limitations of the study

The metabolome is affected by several internal factors such as gene and protein activity and external factors, including diet and environmental factors [15]. In our study, privately owned dogs were used, which were housed and fed under different conditions. Furthermore, 21 breeds and dogs aged between 2 and 8 years were included. To what extent these heterogeneous conditions may have affected the results remains open. However, the included dogs may better reflect the real-life heterogeneity seen in pregnant dogs than laboratory animals.

Experienced breeders kept most dogs so that appropriate housing and feeding under consideration of the reproductive status of the female dog is likely.

A further limitation is that dogs did not fast before sampling because this would not have been acceptable for pregnant and lactating dogs. The timing of collection of the blood samples also showed a mild variation of a few hours; therefore, the impact of diurnal influence cannot be excluded. However, as they were usually sampled in the morning, diurnal variance would be expected to be small. Another potential limitation is the small number of dogs originating from one geographical area.

Some comparisons of our findings with findings from human pregnancies need to be interpreted with caution. Pregnancy of dogs is around 63 days, significantly shorter, and the offspring usually is much larger. Puppies are born physiologically immature and dependent on their mother´s care and milk supply [57]. In the study, the dogs delivered between 1 and 12 puppies.

In addition, research on pregnant dogs has led to several contradictory findings, which makes interpretation of actual metabolic and hematologic findings difficult [1]. For example, it was described more than 40 years ago that some bitches develop progressive, normochromic, and normocytic anemia beginning in mid-pregnancy [9]. While some authors consider this normal, other groups suggest that haematocrits should remain in the normal reference range and that bitches with anaemia should be examined for other concurrent diseases [58].

The samples for this study were taken throughout the year. However, previous research using the same method has shown that only minimal seasonal variation exists in the vast majority of metabolites [59]. Therefore, the season was not taken into account during the data analysis.

In our current study, samples were stored at −80°C for up to 30 months before analysis. In a previous study validating the NMR platform, all studied metabolites were stable at −80°C for at least 12 months [14]. However, the impact of more extended storage on canine samples is lacking.

In many species, it can be challenging to determine whether changes detected during pregnancy are related to the presence of a conceptus or the increased progesterone concentrations [39].

## Conclusions

Several metabolites such as albumin, GlycA, fatty acids, lipoproteins, glucose, and some amino acids show similar patterns during canine pregnancy and lactation as shown in humans. The patterns of some other parameters, such as branched-chain amino acids, alanine and histidine, seem to differ.

To date, it is unclear for most metabolites if observed changes arise from modified resorption from the intestines, modified production or metabolism in the maternal tissues, or their migration through the placenta. In addition, it has hardly been investigated so far to what degree fetal fluid concentrations of specific metabolites are influenced by maternal concentrations [7]. Hence, further species-specific research is needed to shed more light on these questions.

Understanding the molecular changes during pregnancy and the relation to discrete biological events such as embryo development and placentation may support understanding of these processes, learn more about causes of infertility or diseases, and aid in identifying treatments for these conditions [59]. It is essential to understand physiological changes during pregnancy and lactation so that these changes are not misinterpreted as indicators of disease or malnutrition. Furthermore, knowledge of the physiological changes is important for designing studies

on metabolites to adjust inclusion criteria, i. e. if or not to include also pregnant or lactating bitches, and for interpreting the results.

Further research should provide molecular reference ranges in normal pregnancies in relation to studies of adverse pregnancy outcomes [3]. In that regard, metabolomics findings may help monitor vital functions of pre-, intra-, and post-partum bitches and early detection of illness [1].

## Supporting information

**S1 File. Metabolomics raw data.**
(CSV)

## Acknowledgments

The authors wish to thank the participating breeders for their cooperation and interest in the study.

## Author Contributions

**Conceptualization:** Sebastian P. Arlt.

**Data curation:** Sebastian P. Arlt, Claudia Ottka, Hannes Lohi, Janna Hinderer, Julia Lüdeke.

**Formal analysis:** Sebastian P. Arlt, Alexander Bartel.

**Investigation:** Sebastian P. Arlt, Claudia Ottka, Hannes Lohi, Janna Hinderer, Julia Lüdeke, Elisabeth Müller, Corinna Weber, Barbara Kohn.

**Methodology:** Sebastian P. Arlt, Claudia Ottka, Hannes Lohi, Alexander Bartel.

**Project administration:** Sebastian P. Arlt.

**Resources:** Sebastian P. Arlt, Hannes Lohi, Elisabeth Müller, Corinna Weber.

**Supervision:** Sebastian P. Arlt.

**Validation:** Sebastian P. Arlt, Claudia Ottka.

**Visualization:** Alexander Bartel.

**Writing – original draft:** Sebastian P. Arlt, Alexander Bartel.

**Writing – review & editing:** Sebastian P. Arlt, Claudia Ottka, Hannes Lohi, Janna Hinderer, Julia Lüdeke, Elisabeth Müller, Corinna Weber, Barbara Kohn, Alexander Bartel.

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
