## [Decision Letter · Decision Letter 0]

1 Feb 2023

PONE-D-22-35093Metabolomics during canine pregnancy and lactationPLOS ONE

Dear Dr. Bartel,

Thank you for submitting your manuscript to PLOS ONE. After careful consideration, we feel that it has merit but does not fully meet PLOS ONE’s publication criteria as it currently stands. Therefore, we invite you to submit a revised version of the manuscript that addresses the points raised during the review process.

We look forward to receiving your revised manuscript.

Kind regards,

Mükremin Ölmez

Academic Editor

PLOS ONE

Journal Requirements:

https://helda.helsinki.fi/bitstream/handle/10138/335051/vcp.12954.pdf?isAllowed=y&sequence=1

https://academic.oup.com/ajcn/article/111/2/351/5638892?login=false

https://daneshyari.com/article/preview/8403911.pdf 

In your revision ensure you cite all your sources (including your own works), and quote or rephrase any duplicated text outside the methods section. Further consideration is dependent on these concerns being addressed.

"The author(s) received no specific funding for this work. The costs were covered by the Freie Universitaet Berlin (examoination, sampling) and PetBiomics Ltd (Analyses)."

"I have read the journal's policy and the authors of this manuscript have the following competing interests: Claudia Ottka was an employee and Hannes Lohi is a shareholder and the chairman of the board of PetBIOMICS Ltd, who developed and provides the metabolomics test. All other authors declare to have no competing interests."

We note that you received funding from a commercial source: PetBIOMICS Ltd

Additional Editor Comments:

Dear Alexander Bartel,

Your manuscript has now been reviewed by experts in the field. Please find your manuscript with the referee reports in the attachments. Regards.

Reviewers' comments:

Reviewer's Responses to Questions

**Comments to the Author**

1. Is the manuscript technically sound, and do the data support the conclusions?

Reviewer #1: Yes

Reviewer #2: Yes

2. Has the statistical analysis been performed appropriately and rigorously? 

Reviewer #1: Yes

Reviewer #2: No

3. Have the authors made all data underlying the findings in their manuscript fully available?

Reviewer #1: Yes

Reviewer #2: Yes

4. Is the manuscript presented in an intelligible fashion and written in standard English?

Reviewer #1: Yes

Reviewer #2: Yes

5. Review Comments to the Author

Reviewer #1: The manuscript (Metabolomics during canine pregnancy and lactation) by Alexander Bartel is interesting for the researchers and scientific community. Overall, this is a clear, concise, and well-written manuscript. Therefore, I must recommend this for publication after following minor revisions.

Reviewer #2: Dear authors,

Studies on the metabolic profile of dogs at various stages of pregnancy and lactation are very limited. For this reason, the study is original and contains very high quality data. But I recommend that you review the writing of almost all parts of the work.

Corrections to the manuscript are shown in the attached pdf file.

6. PLOS authors have the option to publish the peer review history of their article (what does this mean?). If published, this will include your full peer review and any attached files.

Reviewer #1: No

Reviewer #2: No

---

## [Author Response · Author response to Decision Letter 0]

16 Mar 2023

Reviewer #1: 

The manuscript (Metabolomics during canine pregnancy and lactation) by Alexander Bartel is interesting for the researchers and scientific community. Overall, this is a clear, concise, and well-written manuscript. Therefore, I must recommend this for publication after following minor revisions.

Page 1: choose Key words that are more relevant

 AU: changed

Line 22-42: I suggest major re-write of the abstract including results in precise manner

AU: Abstract has been re-written. Thank you.

Line 44: Improve the Introduction part of the manuscript with little more background.

AU: Some more details on metabolism demands have been included (lines 52 – 58)

Line 68: In line 68 please replace and with ,

AU: Done

Line 73: You may replace word Dog with Bitch or Female Dog

AU: Replaced with female dog.

Line 77: in line 77 please correct some grammatical errors

AU: Corrected.

Line 80: Instead of word dogs use Bitch, it will be more appropriate

AU: Done

Line 88: please mention abbreviation "P" stand for

AU: Added

Line 108: In line 108 NMR stands for ? avoid use of abbreviation first time in the manuscript

AU: Nuclear magnetic resonance (NMR) added.

Line 110: in line 110 please add space

AU: We tried to find words or other parts of the text where an additional space is necessary. Sadly, we were not able to find any positions that we believed were meant by this comment. Please, could you indicate in more detail where you see the need of another space? 

Line 389: Please update references

References: 

Ellen Behrend,2018 some other latest references are available

AU: Reference has been added, thank you

Sebastian P.Arlt 2020 latest references are available.

AU: More recent references have been included

please check Jiaomei Yang 2022 reference

AU: Has been included in the introduction

Please use some latest reference. Concannon,1989

AU: Another and more recent reference has been added

There are more latest references available please update references. Lain, 2007

AU: A more recent reference has been added

 

Reviewer #2: 

Dear authors, Studies on the metabolic profile of dogs at various stages of pregnancy and lactation are very limited. For this reason, the study is original and contains very high quality data. But I recommend that you review the writing of almost all parts of the work.

Line 23-31: Dear authors, Please revise the Abstract section. When I read the Abstract, I don't fully understand your study. So write briefly the material, method and results of your study. Please do not write "Introduction section" descriptions in this section.

AU: Thank you, we revised the abstract intensively

Line 46-48: Please, could you revise this sentence?

AU: revised

Line 52: ??? Which parameters?

AU: specified

Line 60: The main purpose of this study is actually this sentence. Therefore, please explain why some parameter changes are important in the relevant processes.

AU: We have revised the beginning of the introduction according to the suggestions by reviewer 1. We hope that in this context this sentence is more clear now (now line 78 ff).

Line 65: [9,10]

AU: fixed

Line 70: One of the most important factors affecting the metabolic profile is diet. There is no information about this in the material and method section. Dogs of different owners cannot be fed the same diet. How did you eliminate this?

AU: We have added a sentence in the materials and methods section (line 89). Thank you! 

We agree that the heterogeneity of feeding plays an important role. Therefore, we discussed this (see limitations of the study). Feeding a standardized diet would have resulted in a very low compliance of the owners. Furthermore, we included dogs of 21 breeds, which is also a significant confounder. Since we tested every dog at six time points, we use every dog as its “own control”, which might significantly reduce the confounding effects of the different attributes. 

Line 72: Using animals with similar characteristics in metabolic profile studies always provides more reliable results. Isn't this important to you? Why didn't you use dogs with similar characteristics in your study? The number of dogs you use and the characteristics of the dogs are not enough, so it means that more comprehensive studies should be done on this subject. If there is a need for other studies, why didn't you do such a more comprehensive study?

AU: We agree that studies under much more standardized conditions, e. g. in a uniform Beagle colony may provide much more homogeneity (internal validity). However, these studies do not reflect the heterogenous situations we see in veterinary practice. Furthermore, a uniform colony may suffer from unknown individual features which might limit the external validity of findings. 

To reduce bias, we made sure that we really have samples from all six time points in the study so that no samples from different dogs at different time points were compared. This approach may still provide deep insights into the changes of metabolites since every dog served as its “own control”. We just assume, that the changes over the course of the pregnancy are similar between dogs (within animal) and not that dogs of different body weights have similar metabolite levels. 

Line 72: What kind of diets were the dogs fed before pregnancy, during pregnancy and lactation? You should explain about it. The content of the diet may have been effective on serum metabolic profile markers. Please provide detailed information about it.

AU: As written above and thanks to your comments, we included a statement that all dogs were fed individual diets chosen by the owner. In general, based on common sense we advised to feed the normal diet the dog was used to during pregnancy. In the last third we suggested an increase of food supply to 120% (small litter) or up to 150% (large litter). Most breeders began to mix puppy food under the ration from around ten days before parturition (increasing proportions). After parturition the food supply was further increased until week 3 (up to 300% for dogs with very large litters). In all these cases the owners chose their favorite food for their dog and adjusted it to the stage and to the litter size.

Line 72-73: Age, parity, body weight and breed (perhaps) are effective on metabolic profile. How did you eliminate relevant factors in your study?

AU: Again, we agree that these are important factors. But since every dog can be seen as its “own control”, confounding effects are limited. From all dogs samples were taken at every time point. 

Line 80: “80 Dogs were initially presented for ovulation timing.” Please explain. 

AU: Thank you for pointing this out. This was a mistake and not the case in this study. We have deleted this information

Line 80: “After they met the inclusion criteria,” Which criteria?

AU: Inclusion criteria were given in line 84ff. We enrolled only dogs which were healthy, receiving no medication and becoming pregnant during the project.

Line 84: Isn't it a great shortcoming not to collect blood on delivery day?

AU: We agree that on day of delivery we might have revealed special metabolic effects because of uterine labor, onset of lactation and other phenomena. However, since we have used privately owned dogs this was not possible. Stress is a major cause for dystocia and we wanted to avoid any intervention that may have been negative in terms of a normal parturition, potentially leading to C-section. In addition, the owner’s compliance for blood samples during or shortly after delivery would have been poor. 

Line 96: Did you include post-cesarean delivery cases in the study? Such cases are unacceptable. You identify physiological changes. Cesarean delivery cases can change your values.

AU: Thank you for this important remark. Indeed, three dogs underwent an emergency C-section (no planned/elective ones). All dogs did not have C-sections before the beginning of the study. All three dogs having C-sections in the course of this study recovered very fast so that we assume that the surgery did not affect the metabolomic features around three weeks after parturition (L). But we have now indicated the Cesareans in Table 1. We suggest keeping these animals. 

Line 120: remove

AU: We suggest keeping this sentence in order to guide interested readers to more information about other uses of nuclear magnetic resonance metabolomics, since this method is not widely known yet. 

Line 151: Can you add the averages of parameters such as body weight, age, parity at the bottom of the table? please.

AU: Have been added

Line 195: Can you add the studies on the change of fatty acids in dogs during pregnancy and lactation? Please include results from dog references alongside human references.

AU: reference has been added (line 217)

Line 196: Yes, I agree, but this sentence needs a little more clarification. Because if the relationship between fatty acids and immune functions is not explained in a few sentences, it creates confusion. Immune cell activation, relationship with T cells, antigen presentation etc.

AU: We have added two sentences to explain better the link between metabolism and the immune system (line 220 – 225)

Line 199: This is a good result. What might have caused this increase? Can you add a description? 

AU: The reasons for the increase of cholesterol is not clear for us. Like you, we usually would have expected a later rise (if any) in the phases of higher energy demands. 

Line 200: I think there is a tendency for total cholesterol to increased in late pregnancy. We can see the worst result of not having dogs under similar breeding conditions, age and weight in the lipid profile.

AU: all cholesterol parameters follow the same pattern with a peak in mid-pregnancy. We guess that future studies should use more sampling time points. Maybe the then found patterns and underlying reasons become more clear.

We suggest that - for the moment - we need to accept that we found this pattern in privately owned dogs which represent the “average” dog population better than a standardized dog colony. But we also agree that these thrilling findings need to be evaluated much more in future projects to learn more about influences such as age, weight etc.

Line 205: increases

AU: changed

Line 209: remove

AU: we suggest keeping this statement because it may help readers not familiar with lipoproteins to better understand the parameters (now line 237)

Line 210: In this study? Or is it the result of other studies?

AU: In this study, clarified. (now line 239)

Line 218: Can you revise this discussion text? There should be studies on dogs.

AU: Thank you. Two other studies are now referenced (line 244)

Line 221: https://academic.oup.com/jn/article/135/8/1960/4663933?login=false How does PUFA change in dogs?

AU: Added a sentence, that PUFA demand is also increased in pregnant dogs, citing the provided paper. Thank you! (line 254)

Line: 226: Should the serum concentration be increased if Omega-3 is needed to ensure the physiological development of nervous tissue, retinal tissue, kidney, liver and skin functions? Shouldn't the amount be reduced as it is used?

AU: These are very interesting questions which we believe cannot be answered, yet. Therefore, we wrote in the conclusions: “To date, it is unclear for most metabolites if observed changes arise from modified resorption from the intestines, modified production or metabolism in the maternal tissues, or their migration through the placenta”. It is indeed of high interest that we learn more about factors enabling homoeostasis or increase/decrease of specific parameters and of course about potential consequences if the demands for specific metabolites are not met. But these questions cannot be answered by our results.

Line 229: DHA is not increased in the last period of pregnancy. Actually, it should have increased in the last period of pregnancy, because it is very important for puppies. You should add mechanisms to the text about it. Why is the concentration decreased and not increased in the last trimester of pregnancy?

AU: Also here we might see the consequences of an increased uptake of DHA by the puppies, leading to decreased concentrations in maternal blood. We need much more specific studies – for example comparing the concentrations in blood serum of the mother with the ones in fetal serum or in amniotic fluid.

We suggest that we just describe these findings because underlying mechanisms are widely unknown.

Line 232: remove

AU: has been removed

Line 235: Why exactly did you measure arachidonic acid? What can this be indicative of in pregnant and lactating dogs. Could you please add such topics to the text? Please revise the rest of the discussion section based on what I wrote above. The discussion "increased/decreased in pregnancy in women, increased/decreased in our study" overshadows the quality of your study. Please include in the text the mechanisms by which the decrease or increase occurred. Include information in the discussion section, with studies of dogs or cats rather than women.

AU: Arachidonic acid was part of the metabolomics measurements and, therefore, included here. We are sorry to read that you find the way of discussion inappropriate. Our intention is to compare our results with other results from dogs. Since not much is known and potentially a comparison with results from other species might help to explain features in dogs and humans, we decided to use the actual approach. Would you suggest that we delete all comments on human metabolomic changes? 

Line 237: AA

AU: has been changed

Line 237: remove

AU: has been removed

Statistical Analyses

Line 130: Why did you use the Friedman-Test? Why didn't you use parametric tests? Can you give a guiding explanation in the text about this? 

AU: The statistical section was amended (lines 144 ff). The Friedman test was chosen because of 2 requirements for the analysis. First, we have a repeated measures problem, since every animal was measured 6 times (i.e. 6 time points). The Friedman Tests allow to account for this, thus comparing the changes in metabolites within each animal over the course of pregnancy. The second problem was the normality assumption. Since we measured 123 metabolites the normality assumption must be checked for every metabolite. This is not reliably possible with the limited number of observations we have. Therefore, a non-parametric test was chosen since it is more robust. 

Why didn't you make a pairwise comparison of sampling days? Please add statistical analysis. Isn't it necessary to make a pairwise comparison of the parameters that Friedman-Test results are statistically significant?

Figure 1: Include pairwise comparisons of sampling days of parameters that multiple comparison is statistically significant.

Figure 2: Include pairwise comparisons of sampling days of parameters that multiple comparison is statistically significant.

AU: In our approach no post-hoc tests are needed. While the ANOVA/Friedman-Test is commonly used as a prerequisite for post-hoc tests, the ANOVA was developed as a fully independent statistical test. The null hypothesis is that there are no changes in metabolite levels within each animal over the course of the pregnancy. The alternative hypothesis being that there are metabolic changes. This is exactly what we wanted to test for. Thus, the test was chosen according to our scientific questions.

Line 192: P = 0.647 I guess there is no trend. There were individual high values in the data set. I don't think this means that there might be a trend.

AU: The effect of total fatty acids with P = 0.647 is a cumulative effect of all included fatty acids in this variable. Therefore, the changes observed for e.g. Docosahexaenoic acid, Arachidonic acid or Omega-3 influence the concentration of total fatty acids. Other fatty acids like Palmitic acid or Stearic acid dilute this effect. The change in total fatty acids is therefore a sum of its parts and is not an independent random variable. We think going from big (cumulative parameters) into small (specific parameters) is a reasonable and logical approach.

Thus, the p-values are not relevant for the biological interpretation of the observed changes. Additionally, p-values are not a measure of trend or effect size, please see the “The American Statistical Association’s Statement on p-Values” (https://doi.org/10.1080/00031305.2016.1154108) and non-significant p-values do not indicate no effect.

 

Line 193: P=????

AU: please see comment below

Line 199: Please give P value

AU: We omitted the p-values in the text, to avoid duplication of information, but more importantly because p-values have a limited interpretation for ‘omics data analysis. Classical statistical analysis assumes that the number of observations is a multiple of the number of independent variables. For high-dimensional data with lots of variables (123 metabolites) this is often not the case. Therefore, we used dimensionality reduction (k-means Clustering) to create 5 independent latent variables. This assumes that within a cluster, observed changes in each metabolite are not independent of each other. This is a common approach in ‘omics data analysis (https://doi.org/10.1093/aje/kwx016, https://doi.org/10.1186/1471-2105-12-253, https://doi.org/10.1186/1471-2105-13-24). We added this aspect to the statistical methods. Thank you for your comments, we realized that most of our goals and assumptions of the statistical analysis were not stated clearly and hope the new version is more transparent.

Figure Legends: Remove 

AU: We suggest to not remove this legend to make it easier for the reader to see the used tests

---

## [Decision Letter · Decision Letter 1]

4 Apr 2023

Metabolomics during canine pregnancy and lactation

PONE-D-22-35093R1

Dear Dr. Bartel,

We’re pleased to inform you that your manuscript has been judged scientifically suitable for publication and will be formally accepted for publication once it meets all outstanding technical requirements.

Kind regards,

Mükremin Ölmez

Academic Editor

PLOS ONE

Additional Editor Comments (optional):

Reviewers' comments:

Reviewer's Responses to Questions

**Comments to the Author**

1. If the authors have adequately addressed your comments raised in a previous round of review and you feel that this manuscript is now acceptable for publication, you may indicate that here to bypass the “Comments to the Author” section, enter your conflict of interest statement in the “Confidential to Editor” section, and submit your "Accept" recommendation.

Reviewer #2: All comments have been addressed

2. Is the manuscript technically sound, and do the data support the conclusions?

Reviewer #2: Yes

3. Has the statistical analysis been performed appropriately and rigorously? 

Reviewer #2: Yes

4. Have the authors made all data underlying the findings in their manuscript fully available?

Reviewer #2: Yes

5. Is the manuscript presented in an intelligible fashion and written in standard English?

Reviewer #2: Yes

6. Review Comments to the Author

Reviewer #2: Dear Authors,

I would like to express my disagreement with the use of certain statistical analysis methods in the manuscript. Furthermore, as I have previously noted in my corrections, there seems to be a lack of clear explanations for why certain parameters were analyzed. Perhaps you attempted to measure a series of measurements in dogs, but the purpose of this study remains unclear.

While the recent additions made to the Discussion section are a step in the right direction, I believe there is still room for improvement. Although the manuscript is now more coherent, the explanations for the statistical methods and parameter analyses require further elaboration.

As you have mentioned, conducting a more specific study on dogs would result in more precise data. Therefore, I encourage you to consider conducting a more focused study on this particular species.

Regarding the use of birthday sampling, I understand the importance of measuring metabolic changes from birth onwards. However, it may be more beneficial to consider postpartum measurements in carnivores at an earlier stage, as the metabolic changes may not be as significant during the later stages.

Thank you for your attention to these matters, and I look forward to seeing the progress of your work.

Best regards,

7. PLOS authors have the option to publish the peer review history of their article (what does this mean?). If published, this will include your full peer review and any attached files.

Reviewer #2: No

---

## [Editor Report · Acceptance letter]

12 Apr 2023

PONE-D-22-35093R1 

Metabolomics during canine pregnancy and lactation 

Dear Dr. Bartel:

I'm pleased to inform you that your manuscript has been deemed suitable for publication in PLOS ONE. Congratulations! Your manuscript is now with our production department. 

Kind regards, 

on behalf of

Dr. Mükremin Ölmez 

Academic Editor

PLOS ONE